# Unveiling the influence of social network characteristics on sustainable performance in small and medium-sized enterprises: A dynamic capabilities perspective

Tianfeng Jiang[1], Yumeng Zhang[2]*, Gang Chen[3‡], Haixia Feng[2‡], Hao Gao[2‡], Qian Cui[2‡], Tian Lan[2‡], Yueyang Zhang[2‡]

**1** Business School, Shandong University, Jinan, China, **2** Business School, Qingdao University, Qingdao, China, **3** School of Business Administration, Lanzhou University, Lanzhou, China

☯ These authors contributed equally to this work.
‡ These authors also contributed equally to this work.
* 17854212332@163.com

## Abstract

Gaining a deeper understanding of the mechanism through which social networks influence sustainable performance and exploring the role of resource bricolage in enhancing competitiveness are valuable. These insights can help small and medium-sized enterprises (SMEs) better adapt to market changes in the digital age and ultimately, this can enable SMEs to achieve sustainable development. In the context of accelerated technological change and resource scarcity, the question of how SMEs can leverage network resources to achieve sustainable growth has become a pressing issue. This study, grounded in social network theory and resource-based theory, develops a theoretical model exploring the relationships between social network characteristics, resource bricolage, and SME sustainable performance. Using survey data from 230 SMEs, the study empirically demonstrates that both social network scale and social network strength have significant positive impacts on sustainable performance. Additionally, resource bricolage mediates the effects of social network scale and strength on SME performance. Moreover, opportunity identification and opportunity exploitation-key dimensions of dynamic capabilities—moderate the relationship between resource bricolage and sustainable performance. This study highlights the mechanisms through which SMEs can leverage social networks to enhance their sustainable performance, offering theoretical guidance for effectively identifying and utilizing network resources in uncertain and dynamic environments.

## Introduction

Social networks play a crucial role in the modern business operations of enterprises. Through social networks, businesses can establish deep interactions with customers,

**Data availability statement:** This study employed a questionnaire-based survey to collect respondents' demographic information and their perspectives on corporate sustainability performance and related aspects. The instrument incorporated inquiries into personal details including gender, age, income, and other private information, structured in compliance with research ethics protocols and privacy regulations. Consequently, all collected data will be strictly maintained as confidential records without public disclosure. The ethics declaration documentation has been submitted as supplementary materials under the designation "Other", with the file named Research Ethics Review Certification Materials. The Research Ethics Office of the School is not set up separately, and the school will be responsible for research ethics. The contact information is: 85953587. The Office of Ethics (http://ibc.qdu.edu.cn/xygk/zzjg.htm) can be reached at qdsxydw@163.com.

**Funding:** This paper is supported by the National Social Science Foundation's Key Project. 23AGL001, which was received by GH and LT.

**Competing interests:** The authors have declared that no competing interests exist.

partners, and employees, gain insights into market demands and feedback, identify new partners, and collaborate on market expansion or innovation. The diversified information exchange platforms provided by social networks facilitate the acquisition and sharing of knowledge and information, thereby driving innovation. However, social networks also present certain challenges and risks, such as the rapid spread of information, the ease with which negative information can proliferate, and the difficulty in managing online reputations.The healthy development of enterprises as an effective lever for driving the transformation and upgrading of economic structure. To achieve this goal, pursuing high-tech and high value-added industries is undoubtedly essential. However, issues such as deviations in enterprise integration concepts, insufficient innovation drive, low levels of integration among key plays, limited military-civilian technology transfer, and inadequate innovation resources significantly hider the long-term development of enterprises [1], These challenges undermine the industrial foundation necessary to support enterprise growth. Consequently, enterprises experience phenomena such as "bubblelization" and "homogenization", with a severe decline in the interactive effects among stakeholders. In order to mitigate the negative impact of resources scarcity on the enterprise growth, it is crucial for enterprises to explore rational pathway for resource acquisition.

In recent years, academic research on enterprise development performance have deepened, covering topics such as enterprise concepts, development status, and evolutionary processes [2,3], business development models and performance evaluation [4,5], corporate financing and resource allocation [6,7] and the efficiency of technology transfer and its influencing factors [8–10]. However, empirical research on the mechanism and pathways through which enterprise performance operates remain insufficient. Given the significant role that enterprise development plays in promoting national defense economic growth, maintaining social stability, and expanding employment, it is essential to explore the factors influencing enterprise performance within the context of China.

Technological barriers often hinder enterprises' technological development due to limited innovation resources. These resources evolve rapidly, with some depreciating quickly or becoming non-renewable. As a result, enterprises face inefficiencies and resource waste, which further constrain their innovation capabilities [11]. The key to overcoming these challenges lies in collaborating with other innovation partners and the acquisition of resources for innovative development [12]. In the field of entrepreneurship, existing studies employs diverse methodologies, including case studies, meta-analysis, and empirical analysis based on primary or secondary data [13] to examine the impact of social network on enterprise growth [14], organizational behavior [15], and performance [16]. These studies also explore the effect of factors such as emotional trust [17], environmental uncertainty [18] and other complex entrepreneurial scenarios on capabilities like knowledge integration, resource acquisition, risk perception, all of which influence enterprise development.

In summary, extensive social networks enable enterprises to access more resources, with businesses positioning themselves as the central node within these networks to achieve mutual benefits through resource sharing, information exchange,

and technology integration. However, some scholars have raised concerns, suggesting that the expansion in the scale of social network may, to some content, affect the network's stability, potentially having adverse effects on the enterprises performance and development. The close ties between enterprises and other social entities make networks more tightly interconnected, thus reducing cooperation risks. Yet, the question remains whether such close communication could burden enterprises, which could negatively impact performance and development. In the context of new challenges, the influence of strong and weak network relationship on enterprise performance warrants further investigation. Based on these issues, this study asserts the necessity of examining whether and how the characteristics of social network affect enterprise performance.

Resource bricolage is characterized by innovation, flexibility, and the ability to take immediate action. It helps reduce innovation risks and costs while continuously generating new ideas and business opportunities for enterprises. Through resource bricolage, enterprises can break and restructure existing business channels and cooperation networks, creating a more flexible and diverse business structure. This process not only lowers enterprise risk but also enhances adaptability in an ever-changing environment. Most scholars have explored the impact of resource bricolage on enterprise resources, considering it as a mediating variable that influences innovation and development at the micro level. However, researches on its role in the innovation development of mature enterprises remain relatively limited. Enterprises often face resources scarcity, and bricolage behavior facilitates cooperation with other social entities, The impact of this on enterprise performance requires further investigation.

To summary, based on social network theory and resource-based theory, this study develops a "Characteristics-Behavior-Performance" theoretical model. It primarily explores how enterprises, facing resources constraints, can leverage connections with other social entities within their networks to acquire the heterogeneous resources necessary for innovation, thereby uncovering the core mechanism of network structure. In terms of research methodology, case studies and grounded theory typically focus on investigating a single enterprise or a small number of enterprises, and the generalization of the finding may be limited. Therefore, this study will select representative enterprises from different regions of China --- such as the eastern, central, and western regions --- and conduct a detailed investigation of their innovation performance at the micro level. This approach enriches the development of entrepreneurship theory.

From the perspective of current research, the research on social network and enterprise performance is relatively abundant [19]. However, due to the plight of small and medium-sized enterprises (SMEs) facing threats to their survival due to insufficient resources [20], there is still a gap in the research on social network and sustainable performance of SMEs with SMEs as the main body of research. In addition, from the perspective of resource patchwork, the existing literature has not been answered to explore the mechanism of social network and sustainable performance.The contributions of this study are as follows: Based on resource-based theory and social network theory, this research explores the relationship between social network structure, resource bricolage, and sustainable performance. It also tests the boundary effects of dynamic capabilities on the relationship between resource bricolage and sustainable performance, thereby expanding the theoretical applicability in different contexts. This study strongly aligns with the perspective proposed by scholar Wang [21], which emphasizes the importance of network resources in sustaining a firm's competitive advantage.

The rest of the paper is organized as follows. Section "Theoretical Background and Hypotheses Development" provides a theoretical analysis and proposes research hypotheses. Section "Methodology" introduces the source of sample data, the measurement process of variables, etc., and analyzes the data results in detail. Section "Discussion and Conclusion" discusses the research findings and offers managerial implications.

## Theoretical background and hypotheses development

Our study, based on social network theory and resource-based theory, develops a theoretical model linking social network characteristics with sustainable performance. It also tests the mediating role of resource bricolage and the moderating effect of dynamic capabilities. In the subsequent sections, we discuss the proposed theoretical hypotheses.

 

## Social network characteristics and enterprise sustainable performance

Research on the characteristics of enterprise social network is a core direction in the study of innovation networks in businesses [22]. Existing studies have analyzed social networks from various perspectives, including network size, relationship strength, centrality, and structural holes [23]. As enterprises integrate modern technologies, they exhibit new characteristics in terms of resource needs, institutional inertia, and development pathways. The size and strength of social networks are closely related to resource acquisition. Based on this, this study will explore the size and strength of social networks.

The size of social network refers to the number of other social entities with which an enterprises engages in business activities, reflecting the number of network members and the complexity of their connection [24]. Based on the resource-based theory, enterprises with a short development history often face challenges such as limited funding, insufficient management skills, and lack of market experience [25,26], making it difficult to achieve innovative project development relying solely on existing resources. Through social network relationships, enterprises establish collaborative ties with the entities such as governments, universities, research institutions, other businesses, and the military, coordinating and integrating internal and external innovation resources. In the process of interacting with these entities, enterprises' advanced and cutting-edge innovation ideas and development strategies collide with previously closed and outdated concepts, resulting in updates and upgrades. This enables innovative products to better meet the actual needs of diverse customers, solidify market positions, and improve enterprises performance.

Hence, we proposed the following research hypothesis

H1a: The size of a social network has a positive impact on an enterprise's sustainable performance.

H1b: The sterngth of a social network has a positive impact on an enterprise's performance.

## Resource bricolage and enterprise sustainable performance

Resource bricolage is a behavior capability that highly integrates innovation, proactivity, spontaneity, and environmental agency. It plays an indispensable role in improving enterprise performance and fostering development. Through a novel and distinctive perspective combined with rigorous macro-level logic, resource bricolage identifies the attributes and value of an organization's existing resources. By restructuring and integrating these resources, organizations can gain resource advantages under sub-optimal development conditions, mitigate resource constraints, broaden access to resources, reduce expenditure, enhance investment in innovation, and improve innovation outcomes.

According to the resource-based theory, the basis for achieving competitive advantage lies in possessing scarce, valuable, imitable and non-substitutable heterogeneous resources. Under conditions of resource constraints and environmental uncertainty, resource bricolage significantly determines the future development trajectory of organizationst. Enterprises often lack production experience and retain their original business habits, unable to realize the sharing and integration of resources. For example, certain resources may be redundant in one domain while being scarce in another. Similar challenges exist with resources such as knowledge, production technologies, human capital, funding and education. By increasing the occurrence of resource bricolage behaviors, organizations can reorganize, classify, and redesign various resources. This allows them to reconfigure and integrate new resources into different production domains, thereby transcending the boundaries of product innovation. Resources possess multiple attributes, and their utility varies across different innovation situations. Organizations often find it challenging to accomplish tasks such as project development and product innovation relying solely on existing resources. Resource bricolage enables enterprises to reevaluate their current resources, identify new attributes, and reduce dependence on specific existing resources. This process facilitates the integration and innovate use of heterogeneous resources to meet diverse demand [27].

Hence, we proposed the following research hypothesis

H2: Resource bricolage positively influences an organization's sustainable performance

## The mediating effect of resource bricolage

From the perspective of resource-based theory and social network theory, an enterprise is a collection of resources, and the diverse and heterogeneous resources embedded within social networks serving as the foundation for product innovation and sustainable management. The expansion of a social networks signifies an increase in the number of innovative actors with whom the enterprises establishes business connections, thereby enhancing the organization's innovation capacity and garnering multi-faceted support for its developmental efforts.

Social network centered around the enterprise unique characteristics in terms of the selection of collaborative partners. Typically, the operating environment of an enterprise is relatively closed, with higher confidentiality surrounding resources such as technology and knowledge. The existence of technological barriers creates an uneven flow of resources. However, the larger the social network, the more it facilitates behaviors such as knowledge spillovers and external learning, thereby enriching and diversifying the available resources. Consequently, enterprises are increasingly required to engage in resource bricolage, which involves classifying and specializing resources, fully tapping into and recycling redundant attributes. Through reconfiguration, valuable and scarce heterogeneous resources can be generated, thereby alleviating resource shortages faced by the enterprise. Resource bricolage effectively integrates and reconstructs the demand resources within social networks, lending to more efficient resources utilization, reduced resource acquisition costs, and fostering innovation. Ultimately, this process helps enterprises strengthen their competitive advantage in the market, thereby improving overall performance.

Frequent interactions and collaboration between enterprises and other societal subjects can enhance the likelihood of resource-sharing behaviors. Increased trust helps eliminate defensive and opportunistic mindsets, fosters greater sincerity in cooperation, reduces information asymmetry, and strengthens emotional commitment between partners. High level of commitment within social network relationship can significantly enhance the willingness of innovative subjects to collaborate [28]. Enterprises often seek to obtain essential resources through close social ties to address imbalance in innovation resource allocation and industry monopolization. This demand strongly stimulates enterprises to integrate and reconfigure existing resources, thereby upgrading their innovative behaviors. Effective resource bricolage enables enterprises to undertake larger-scale project development and fulfill national responsibilities, enhancing their social reputation. This, in turn, reduces barriers to future business expansion and operational development, ultimately contributing to improved enterprise performance [29].

Hence, we proposed the following research hypothesis

H3a: Resource bricolage mediates the relationship between social network size and and organization's sustainable performance.

H3b: Resource bricolage mediates the relationship between social network strength and an organization's sustainable performance.

## The moderating role of dynamic capability in the relationships between resource bricolage and organizational sustainable performance

Based on the resource-based theory, when enterprises possess robust entrepreneurial dynamic capability, they are better equipped to reassess the value of resources with a forward-looking perspective and applying these resources effectively across various projects. Furthermore, entrepreneurial dynamic capabilities strengthen an organization's ability to detect and respond to external environmental fluctuations [30]. By integrating strategic planning capabilities, organizations can effectively adapt to environmental changes and quickly adjust their production and operational strategies. These capabilities encompass two key aspects: the ability to effectively identify opportunities and the ability to exploit those opportunities.

At different stages of growth, enterprises face varying resource demands and strategic interests. To align with their developmental needs and adapt to dynamic external environments, enterprises must implement corresponding adjustments. Entrepreneurial dynamic capabilities provide a proactive advantage for navigating hyper-competitive environment. Given the complex and volatile market environment, coupled with insufficient innovation capabilities, many enterprises struggle to accurately manage their operational environments. Entrepreneurial dynamic capabilities enable the integration, reconfiguration, and upgrading of internal and external resources, allowing organizations to adjust production and operation plan in response to resources and environmental fluctuations. These capabilities maximize resource advantages, minimize the innovation risks, and support resource acquisition and the identification of opportunities for innovative development [31]. Consequences, these capabilities enhance organizational competitiveness by identifying critical resources, developing innovative products, and increasing resource bricolage behavior. Furthermore, they facilitate the transfer and commercialization of innovation outcomes within complex social network relations.

Hence, we proposed the following research hypothesis

H4a: Opportunity identification capability can significantly moderates the relationship between resource bricolage and organizational sustainable performance.

H4b: Opportunity exploitation capability significantly moderates the relationship between resource bricolage and organizational sustainable performance.

To sum up, the theoretical framework of this study is shown in Fig 1.

## Methodology

### Sample selection

Our study selected a sample of 230 representative small and medium-sized enterprises (SMEs) from various industries, including food and biopharmaceuticals (e.g., Guyu Biotechnology Co., Ltd.), metallurgical machinery (e.g., Hengan Intelligent Control Technology Co., Ltd.), materials (e.g., New Material Technology Co., Ltd.), and new energy (e.g.,f Fengfa

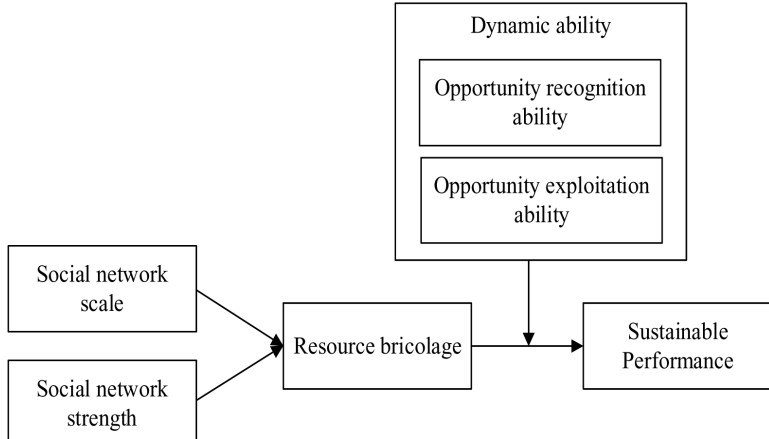

**Fig 1. Theoretical model framework.** (A) On the left side of the diagram are the independent variables, representing the characteristics of social networks, which include two dimensions: social network scale and social network strength. (B) The middle section depicts the mediating variable, representing the degree of resource bricolage. (C) On the right side are the dependent variables, representing organizational sustainable performance. (D) At the top of the diagram are the moderating variables, representing dynamic abilitiy, which consist of two dimensions: opportunity recognition ability and opportunity exploitation ability.

Technology Development Co., Ltd.). Since their establishment, these companies has experienced challenges such as resource shortage and environmental volatility. Through establishing strong networks with other external stakeholders, they have achieved significant scientific technological innovation outcomes. The selected sample reflects a high level of representativeness.

## Data collection and sample overview

Before the survey, the research group explained the contents and objetctives of the survey to the companies. After obtaining their consent, the research group distributed questionnaires to the senior executives and staff from the performance departments of the participating companies. The questionnaire consists of two parts. The first part focus on demographic information, including gender, age, educational background, enterprise size and years of establishment, and its influence on enterprise performance. The second part examed variables such as the characteristics of enterprise social network, resource bricolage, sustainable performance and dynamic capability (See Table 1 for details on variables). A total of 230 companies participated in the survey, Out of 300 questionnaires distributed, 220 were retrieved, and the response rate reached 73.3%.

To account for the potential effects of employee gender, age, educational background, company size and years of establishment on corporate performance, these factors were included as control variables in this study. The finding revealed an approximately equal gender distribution among respondents, with a male to female ratio of nearly 1:1. About 45% of respondents were ages between 20 and 30 years old, while 46.4% were aged between 31 and 40 years. Indicating that the workforce was predominantly young to middle-aged and exhibited strong innovation potential. Nearly 80% of employees held a bachelor's degree or higher, reflecting a high level of educational attainment, which may influence

Table 1. Demographic description of samples.

| Variable | Feature | Number | Proportion(%) |
|---|---|---|---|
| Gender | Male | 125 | 56.8 |
| | Female | 95 | 43.2 |
| Age | <=20 | 0 | 0 |
| | 21-30 | 98 | 44.5 |
| | 31-40 | 102 | 46.4 |
| | 41-50 | 10 | 4.5 |
| | 51-60 | 8 | 3.6 |
| | Over 60 years old | 2 | 1 |
| Education | Primary school | 4 | 1.8 |
| | Junior high school | 15 | 6.9 |
| | Senior high school | 26 | 11.8 |
| | Undergraduate course | 167 | 75.9 |
| | Postgraduate degree | 8 | 3.6 |
| Enterprise scale | <=50 | 55 | 25 |
| | 51-200 | 105 | 47.7 |
| | 201-500 | 49 | 22.3 |
| | 501 and above | 11 | 5 |
| Age of establishment of enterprise | Less than 5 years | 104 | 47.3 |
| | 5 to 8 years | 61 | 27.7 |
| | 8-10 years | 48 | 21.8 |
| | More than 10 years | 7 | 3.2 |

corporate performance. Around half of the respondents were employed in companies with 51–200 employees, representing large organizations capable of fulfilling national and societal responsibilities. Additionally, More than 50% of respondents worked in companies that has been established for over 5 years, with some having operated for 8–10 years or longer. Longer establishment periods often correlate with richer management and operational experience, positively contributing to corporate performance. As shown in Table 1.

## Measurement

Considering the contextual specificity of enterprise, the study modifies the measurement of key variables based on well-established scale from international literature. Pre-testing is conducted to refine the descriptions and language of the scale items, ensuring reliability and validity, thereby enhancing the applicability of the research findings. The questionnaire employs a seven-point Likert scale for responses, with specific details as follows.

Social Network Characteristics: Based on the views of Watson [32] and Tan [33], as well as the work of Luo and Park et al. [34], two representative dimensions, social network scale and social network strength are selected for measurement. The scale of social network consists of five items, with a Cronbach's α of 0.916. The social network strength was measured using six items, with a Cronbach's α of 0.911. Experts generally consider internal reliability insufficient if Cronbach's α is below 0.6, acceptable when it falls between 0.7 and 0.8, and strong when it ranges from 0.8 to 0.9. These resuts indicates a high reliability for the measurement of social network characteristics.

Resource Bricolage: by referring to the research of Davidsson et al. [35], the resource bricolage scale includes six items that assess the integration of existing resources to address with challenges and leverage opportunities to solve new problems. The scale achieves a Cronbach's α of 0.915, indicating high reliability.

Entrepreneurial Dyamic Capability: Based on the research of Ma et al. [36] and Foss et al. [37], focusing on opportunity recognition capability encompass the ability to quickly gather entrepreneurial opportunity information and identify the impact of new information, with a scale reliability of Cronbach's α Of 0.891. Opportunity exploitation ability includes six aspects, such as the ability to develop new products and services, the ability to develop new market areas, with a Cronbach's α of 0.895. Both the opportunity recognition and exploitation capability demonstrate good reliability.

Sustainable Performance: Based on the research of Zhu and Fei [38], to measure sustainable performance, which mainly includes enterprise growth performance and profitability performance. Items include metrics such as employees growth and new products and services growth. The Cronbach's α of this scale is 0.928, indicating that it has good reliability.

## Descriptive statistical analysis

In this study, the entrepreneur's gender, age, educational background, enterprise age and enterprise size were included as control variables. Descriptive statistical analysis of all variables was conducted using SPSS22.0, with the results presented in Table 2: Social network scale and resource bricolage ($r = 0.468$, $p < 0.01$), social network intensity and resource bricolage ($r = 0.442$, $p < 0.01$), social network scale and enterprise performance ($r = 0.521$, $p < 0.01$), social network intensity and enterprise performance ($r = 0.437$, $p < 0.01$). resource bricolage was positively correlated with corporate sustainable performance ($r = 0.546$, $p < 0.01$). No significant correlation was observed between opportunity identification ability, opportunity exploitation ability and independent or dependent variables. All correlation values were below 0.6, and the square root values of the diagonal AVE exceeded the inter-variable correlation, indicating no significant multicollinearity issues. These results suggest the data is relatively reliable.

## Reliability and validity analysis

The assessment of overall reliability and internal consistency reliability is critical in questionnaire surveys to ensure the stability of the results. Cronbach's α coefficient serves as a key indicator for measuring reliability. In general, the higher

**Table 2. Variables mean value, standard deviation and correlation coefficient, AVE square root value results.**

| variable | 1 | 2 | 3 | 4 | 5 | 6 | 7 | 8 | 9 | 10 | 11 |
|---|---|---|---|---|---|---|---|---|---|---|---|
| Gender | 1.000 | —— | | | | | | | | | |
| Age | −.134* | 1.000 | —— | | | | | | | | |
| Education | −0.004 | −0.117 | 1.000 | —— | | | | | | | |
| Enterprise scale | −.151* | 0.109 | 0.129 | 1.000 | —— | | | | | | |
| Age of establishment of enterprise | 0.027 | 0.208** | 0.223** | 0.561** | 1.000 | —— | | | | | |
| Social network scale(X1) | 0.026 | 0.097 | 0.068 | 0.180** | 0.162* | 0.825 | —— | | | | |
| Social network strength(X2) | 0.010 | 0.108 | 0.036 | 0.082 | 0.084 | 0.419** | 0.797 | —— | | | |
| Resource bricolage(M) | 0.015 | 0.061 | −0.002 | −.158* | −0.014 | 0.468** | 0.442** | 0.827 | —— | | |
| Corporate sustainable performance (Y) | 0.036 | 0.043 | 0.023 | −0.071 | 0.030 | 0.521** | 0.437** | 0.546** | 0.854 | —— | |
| Opportunity recognition capability (T1) | 0.122 | −0.042 | 0.033 | 0.046 | 0.124 | 0.074 | 0.117 | 0.124 | 0.023 | 0.766 | —— |
| Opportunity exploitation capability (T2) | −.147* | 0.075 | −0.023 | 0.102 | 0.014 | 0.083 | 0.057 | 0.064 | −0.015 | −0.013 | 0.787 |
| Mean | 1.52 | 1.63 | 2.82 | 3.31 | 2.70 | 5.11 | 4.96 | 5.07 | 5.11 | 4.84 | 5.43 |
| SD | 0.50 | 0.65 | 0.59 | 1.16 | 0.91 | 1.11 | 1.07 | 1.08 | 0.92 | 1.36 | 0.84 |

Note: * means P<0.05; ** represents P<0.01; *** means P<0.001, same below.

the Kronbach coefficient, the higher the reliability. In basic research, the reliability should be at least 0.80 to be considered acceptable; In exploratory studies, a confidence level of 0.70 is acceptable. Confidence between 0.70 and 0.98 is considered high confidence, while confidence below 0.35 is considered low confidence and should be considered rejected.

Using SPSS22.0, the Cronbach's α coefficient of the overall questionnaire was calculated as 0.936, and the Cronbach's α value of internal items of each variable was greater than 0.8, indicating that the scale had good overall reliability and internal consistency reliability.

Factor loading of the questionnaire data were analyzed using AMOS 22.0. The analysis revealed that all factor loadings were all greater than 0.5, composite reliability (CR) was greater than 0.8, and the AVE (Average Variance Extracted) was greater than 0.5, indicating that the model had an ideal interpretation of constructs. The detailed results were shown in Table 3.

In this study, confirmatory factor analysis (CFA) was conducted using AMOS22.0 to test the structure validity of the questionnarie Initially, SPSS22.0 was used to calculate the Kaiser-Meyer-Olkin (KMO) value and Bartlett's test of sphericity for the questionnaire population. The results showed that the overall KMO value was 0.929, exceeding the standard threshold of 0.7. Then, single-factor, two-factor, three-factor, four-factor, five-factor and six-factor models were constructed using the principal component analysis method. The fit indices of each model were analyzed to verify the structural validity of the questionnaire. The results showed that compared with other models, the six-factor model proposed in this study exhibited the best fit (CMIN/DF = 1.791, RMSEA (Root Mean Square Error of Approximation) =0.06, SRMR (Standardized Root Mean Square Residual) = 0.054,CFI (comparative fit index) =0.925, GFI (goodness-of-fit index) =0.817, TLI (Tucker-Lewis index) =0.917). All fit indices met the standards.

In general, a CMIN/DF value closer to 1 indicates a better model fits. Smaller RMSEA and SRMR values signify improved fit, while higher CFI, GFI, TLI values reflect better model performance. These results indicate that the six-factor model constructed in this study demonstrates the most favorable fit, supporting the validity of the theoretical model.

In order to further confirm the good structural validity of the scale, CFA test was performed using the maximum likelihood estimation Bootstrap method in MPLUS software. The results included CFI=0.924, TLI=0.917, SRMR=0.062. For sample sizes below 500, the these values indicate a good model fit. Detailed results are presented in Table 4.

Harman single factor analysis of variance was used to assess the presence of common method bias in the questionnaire. First of all, an unrotated factor analysis was performed on all items in the questionnaire items. The results showed that

**Table 3. Scale item, reliability and validity results.**

| Variable | Item | Standardization coefficient | Cronbach's alpha | AVE | CR |
|---|---|---|---|---|---|
| Social network scale(X1) | The number of cooperation and exchanges with universities and research institutions | 0.855 | 0.916 | 0.687 | 0.916 |
| | The number of cooperation and exchanges with military enterprises | 0.783 | | | |
| | The number of cooperative exchanges with other enterprises | 0.811 | | | |
| | Number of exchanges with key suppliers | 0.817 | | | |
| | The number of cooperation and communication with key customers | 0.874 | | | |
| Social network strength(X2) | Frequency of cooperation and exchanges with universities, research institutions, etc | 0.642 | 0.911 | 0.636 | 0.912 |
| | Frequency of cooperation and exchanges with military enterprises | 0.807 | | | |
| | Frequency of cooperation and communication with other enterprises | 0.823 | | | |
| | Frequency of cooperation and communication with key suppliers | 0.811 | | | |
| | Frequency of cooperation and communication with key customers | 0.848 | | | |
| | Frequency of communication with cooperative partners such as government departments, financial institutions and intermediary service clients | 0.837 | | | |
| Resource bricolage(M) | Find the resources to deal with the challenge | 0.853 | 0.915 | 0.684 | 0.915 |
| | Use resources to respond to new problems or opportunities | 0.787 | | | |
| | Combine existing resources to meet challenges | 0.821 | | | |
| | Find workable solutions from existing resources | 0.883 | | | |
| | Combine resources to complete new challenges | 0.786 | | | |
| Opportunity recognition capability(T1) | Can quickly collect information about entrepreneurial opportunities | 0.875 | 0.891 | 0.731 | 0.891 |
| | Quickly identify the impact of new information | 0.848 | | | |
| | Be aware of novel entrepreneurial opportunities | 0.841 | | | |
| Opportunity exploitation capability(T2) | Good ability to develop ideas | 0.79 | 0.895 | 0.587 | 0.895 |
| | Ability to develop new products and services | 0.788 | | | |
| | The ability to develop new market areas | 0.701 | | | |
| | Ability to develop new entrepreneurial opportunities | 0.747 | | | |
| | Ability to digest, absorb and integrate new market opportunities | 0.774 | | | |
| | Able to handle all kinds of difficulties in starting a business | 0.794 | | | |
| Sustainable performance (Y) | Increase in the number of employees | 0.687 | 0.928 | 0.620 | 0.928 |
| | Sales growth | 0.764 | | | |
| | Net income growth | 0.784 | | | |
| | Market share growth | 0.849 | | | |
| | Growth in new products and services | 0.761 | | | |
| | market occupancy | 0.779 | | | |
| | net profit rate | 0.81 | | | |
| | rate of return on investment | 0.851 | | | |

the first principal component accounted for 16.749% of the total variance, which is substantially less than half of the total explained variance of 72.416%. This indicates that common method deviation has a relatively minimal impact on the study.

## Hypothesis testing

This study employed SPSS 22.0 software and the hierarchical regression analysis to test hypotheses H1 through H4. Gender, age, educational background, enterprise size and enterprise age were included as control variables. The results were shown in Table 5.

**Table 4. Results of confirmatory factor analysis.**

| model | $\chi^2$ | df | $\Box\chi^2/$ df | RMSEA | CFI | GFI | TLI | IFI | SRMR |
|---|---|---|---|---|---|---|---|---|---|
| Six-factor model | 859.551 | 480 | 1.791 | 0.06 | 0.925 | 0.817 | 0.917 | 0.925 | 0.054 |
| Five-factor model[1] | 1462.791 | 485 | 3.016 | 0.096 | 0.806 | 0.635 | 0.789 | 0.807 | 0.084 |
| Five-factor model[2] | 1242.637 | 485 | 2.562 | 0.084 | 0.85 | 0.757 | 0.836 | 0.851 | 0.087 |
| Four-factor model | 1845.173 | 489 | 3.773 | 0.113 | 0.731 | 0.598 | 0.709 | 0.733 | 0.102 |
| Three-factor model | 2671.639 | 492 | 5.43 | 0.142 | 0.567 | 0.495 | 0.535 | 0.57 | 0.199 |
| Two-factor model | 1518.304 | 208 | 5.921 | 0.15 | 0.517 | 0.468 | 0.484 | 0.52 | 0.143 |
| Single factor model | 1856.902 | 209 | 6.815 | 0.163 | 0.428 | 0.423 | 0.39 | 0.432 | 0.152 |

Note: Six-factor model: (X1,X2, M, T1, T2, Y); Five-factor model [1]: (X1+X2, M, T1, T2, Y); Five-factor model [2]: (X1, X2, M, T1+T2); Four-factor model: (X1+X2, M, T1+T2, Y);Three-factor model: (X1+X2, M+T1+T2, Y) ;Two-factor model: (X1+X2+M+T1+T2, Y);Single factor model:(X1+X2+M+T1+T2+Y).

**Table 5. Hierarchical regression Analysis.**

| Interpret Variable | Resource Bricolage | | | |
|---|---|---|---|---|
| | M1 | M2 | M3 | M4 |
| **Contral variable** | | | | |
| Gender | 0.014 | −0.020 | 0.003 | −0.019 |
| Age | 0.078 | 0.035 | 0.033 | 0.015 |
| Education | 0.012 | −0.008 | 0.014 | −0.002 |
| Enterprise scale | −0.223** | −0.297*** | −0.249*** | −0.295*** |
| Age of establishment of enterprise | 0.050 | 0.033 | 0.031 | 0.025 |
| **Independent variable (X)** | | | | |
| Social network scale (X1) | | 0.496*** | | 0.363*** |
| Social network strength(X2) | | | 0.469*** | 0.323*** |
| $R^2$ | 0.04 | 0.28 | 0.26 | 0.37 |
| $\Delta R^2$ | 0.02 | 0.26 | 0.24 | 0.34 |
| F | 1.96 | 13.66*** | 12.358*** | 17.25*** |

Note: Data are standard β coefficient values, * means significant at the probability level of $P < 0.1$, ** means significant at the probability level of $P < 0.05$, *** means significant at the probability level of $P < 0.001$. Same with comments below.

First, the main effect were tested as shown in Table 6. In Model M6 (β = 0.538, p < 0.01) and model M7 (β = 0.447, p < 0.01), social network size and social network strength were introduced into the main effect test model after controlling for the variables. The results indicate that both social network size and social network strength positively influence enterprise performance, supporting hypotheses H1a and H1b. Secondly, The study also examined the impact of social network size and social network strength on resource bricolage. The results were significant, with Model M2(β = 0.496, p < 0.01) and Model M3 (β = 0.469, p < 0.01) showing positive effects. Furthermore, resource bricolage on performance (M8,β = 0.526, p < 0.01), revealing a significant positive contribution to performance improvement. These results validate the main effect analysis and confirm hypothesis H2.

Mediating effect test: The mediating effects of resource bricolage were analyzed to understand its role between social network characteristics and corporate sustainable performance. For social network size, the results of Model M9 (β = 0.344, p < 0.01), revealed a significant mediating effect.Comparing M6 and M9, the inclusion of resource bricolage as a mediating variable reduced the direct impact of social network size on corporate sustainable performance from

**Table 6. Hierarchical regression Analysis.**

| Interpreted Variable | corporate sustainable performance | | | | | | |
|---|---|---|---|---|---|---|---|
| | M5 | M6 | M7 | M8 | M9 | M10 | M11 |
| **control variable** | | | | | | | |
| gender | 0.036 | 0.000 | 0.026 | 0.029 | 0.007 | 0.025 | 0.006 |
| age | 0.051 | 0.004 | 0.008 | 0.010 | −0.008 | −0.005 | −0.017 |
| education | 0.018 | −0.004 | 0.019 | 0.011 | −0.002 | 0.013 | 0.002 |
| enterprise scale | −0.121 | −0.201** | −0.146* | −0.004 | −0.099 | −0.046 | −0.118* |
| Age of establishment of enterprise | 0.058 | 0.039 | 0.040 | 0.032 | 0.028 | 0.028 | 0.026 |
| **independent variable** | | | | | | | |
| Social network scale(X1) | | 0.538*** | | | 0.367*** | | 0.324*** |
| Social network strength(X2) | | | 0.447*** | | | 0.259*** | 0.275** |
| **mediating variable** | | | | | | | |
| resource bricolage(M) | | | | 0.526*** | 0.344*** | 0.402*** | 0.188*** |
| $R^2$ | 0.02 | 0.29 | 0.21 | 0.28 | 0.38 | 0.33 | 0.40 |
| $\Delta R^2$ | −0.01 | 0.27 | 0.19 | 0.26 | 0.36 | 0.31 | 0.38 |
| F | 0.65 | 14.49*** | 9.42*** | 13.66*** | 18.15*** | 14.82*** | 17.59*** |

β = 0.538, β = 0.367 and p < 0.001, indicating a partial mediating effect. This result confirms that resource bricolage partially mediates the relationship between social network size and corporate sustainable performance.

Similarly, for social network strength, the results of Model M10 (β = 0.402, p < 0.01), showed a significant mediating role resource bricolage. From the changes of M6 and M9, it can be seen that when resource bricolage is added as the mediating variable, the impact of social network scale on corporate sustainable performance is significantly positive with β = 0.538, β = 0.367 and p < 0.001, indicating that resource bricolage plays a partial mediating role between social network scale and corporate sustainable performance. Similarly, from the changes between M7 and M10, it can be seen that when resource bricolage is added to M10 model as an intermediary variable, the impact of social network strength on enterprise performance decreases significantly from β = 0.447 β = 0.259 (p < 0.001), when resource bricolage was included as a mediator. This finding confirms that resource bricolage also partially mediates the relationship between social network strength and corporate sustainable performance. Together, these results support the theoretical hypotheses H3a and H3b.

To further examine differences in the mediating effects of resource bricolage between social network size and strength, the Process program based on Hayes and the Bootstrap method were employed. With 5000 Bootstrap samples, the results showed that the indirect effect of resource bricolage between social network size and corporate sustainable performance was 0.14, with a 95% confidence interval (CI) of (0.016, 0.263), indicating a significant mediating effect. For social network strength, the indirect effect was 0.116, with a 95% CI of (0.024, 0.207), also demonstrating a significant mediating effect. These results provide additional confirmation of hypotheses H3a and H3b.

To compare the two mediating effects, a mediating contrast model was constructed. The estimated difference between the mediating effects was −0.024, with a 95%CI of (−0.186,0.079). Since the confidence interval includes 0, it indicates that the mediating effects of resource bricolage on the relationships between social network size and strength with corporate sustainable performance are equivalent. This finding suggests that resource bricolage plays an equally important role in mediating the effects of both social network size and strength on corporate sustainable performance.

In order to verify differences in the mediating effects of resource bricolage between social network size and strength, the Process program based on Hayes and Bootstrap method were employed. With 5000 Bootstrap samples, the results were showed in Table 7 that the indirect effect of resource bricolage between social network size and corporate sustainable performance was 0.14, with a 95%CI of (0.016, 0.263), indicating the mediating effect is significant. For social

**Table 7. Comparison results of mediating effect and double mediating effect.**

| Variable | Standard deviation | Lower level | Upper level |
|---|---|---|---|
| X1-M-Y (a) | 0.042 | 0.016 | 0.263 |
| X2-M-Y(b) | 0.05 | 0.024 | 0.207 |
| a-b | 0.063 | −0.186 | 0.079 |

network strength, the indirect effect was 0.116, with a 95%CI of (0.024, 0.207), also demonstrating a significant mediating effect. These results provide additional confirmation of hypotheses H3a and H3b. To compare the two mediating effects, a mediating contrast model was constructed. The estimated difference between the mediating effects was −0.024, with a 95%CI of (−0.186, 0.079). Since the confidence interval included 0, it indicate that the mediating effect of resource bricolage between social network size and strength with corporate sustainable performance are equivalent. The finding suggests that resource bricolage plays an equally important role in mediating the effects of both social network size and strength on corporate sustainable performance.

Adjustment effect test: Io test the adjustment effect model, this study standardizes the variables of resource piecing, opportunity identification ability and opportunity exploitation ability. Interaction terms are constructed between the mediating and moderating variables and incorporated into the regression model. The results indicate that the interaction between resource bricolage and opportunity recognition ability significantly and positively impacts corporate sustainable performance (M13, $\beta = 0.142$, $p < 0.05$), demonstrating a notable moderating effect. Similarly, the interaction between resource bricolage and opportunity exploitation capability (M15, $\beta = 0.211$, $p < 0.01$) also shows a significant positive effect on enterprise performance. This findings suggest that the opportunity identification and exploitation capabilities, as components of entrepreneurial dynamic ability, significantly moderate the relationship between resource bricolage and corporate sustainable performance, supporting hypotheses H4a and H4b. The regression results are presented in Table 8. To further validate the moderating effects, resource bricolage values plus or minus one standard deviation are plotted. The results show that under conditions of high opportunity identification capability, resource bricolage has a stronger positive effect on enterprise performance compared to low opportunity identification capability. Likewise, when opportunity exploitation capability is high, the slope is steeper, indicating a more pronounced positive impact of resource bricolage on enterprise performance. These findings confirm that the adjustment effect aligns with the expected hypotheses.

## Discussion and conclusion

### Research conclusions and theoretical implications

This study, from the perspective of social network theory and resource-based theory, explores the development pathways and influencing factors for enhancing enterprise performance. The findings and theoretical contributions are as follows:

The empirical research uses social network scale and strength as key indicators of social network structure. It finds that an expanded social network scale—reflected by a broader business scope and diverse service targets—increases the demand for resource richness. This, in turn, encourages resource bricolage behaviors. Engaging extensively in resource bricolage allows firms to reduce dependence on existing resources, facilitate the restructuring and upgrading of these resources, and improve innovation efforts. This, in turn, promote the generation of new resources, and helps firms overcome resource deficiencies, ultimately improving their developmental performance. Additionally, as firms strengthen their connection with partners and other social network entities, the depth of these relationship increase, enabling access to heterogeneous and valuable information resources beneficial for firm development. Close connection with social network entities foster emotional trust, reduce collaboration barriers, and increase the frequency of cooperation, resulting in substantial benefits for firms and consequently improving corporate performance. Previous studies have confirmed that the abundant and heterogeneous resources within social networks positively influence corporate sustainable performance.

 

**Table 8. Hierarchical Regression Analysis.**

| Interpreted Variable | Corporate Sustainable Performance | | | |
|---|---|---|---|---|
| | **M12** | **M13** | **M14** | **M15** |
| **Control variable** | | | | |
| Gender | 0.035 | 0.038 | 0.023 | 0.000 |
| Age | 0.006 | −0.001 | 0.012 | −0.016 |
| Education | 0.011 | 0.021 | 0.010 | −0.024 |
| Enterprise scale | −0.002 | −0.021 | 0.003 | 0.031 |
| Age of establishment of enterprise | 0.039 | 0.074 | 0.029 | 0.020 |
| **Independent variable** | | | | |
| Social network scale(X1) | | | | |
| Social network strength(X2) | | | | |
| **Mediating variable(M)** | | | | |
| Resource bricolage(M) | 0.533*** | 0.564*** | 0.530*** | 0.499*** |
| **Moderator Variable(T)** | | | | |
| Opportunity recognition capability(T1) | −0.056 | −0.060 | | |
| Opportunity exploitation capability(T2) | | | −0.051 | 0.001 |
| **Interaction** | | | | |
| Resource bricolage * Opportunity identification capability | | 0.142** | | |
| Resource bricolage * Opportunity exploitation capability | | | | 0.211*** |
| $R^2$ | 0.283 | 0.301 | 0.282 | 0.319 |
| $\Delta R^2$ | 0.259 | 0.274 | 0.258 | 0.293 |
| F | 11.825*** | 11.243*** | 11.797*** | 12.266*** |

Note: Same as Table 5 for hierarchical regression analysis.

Notably, the enterprises examined in this study represent a new type of politically guided organization, characterized by distinct management and development systems.

Dynamic capabilities significantly moderate the relationship between resource bricolage and corporate sustainable performance. Entrepreneurial dynamic ability, which combines opportunity identification and opportunity exploitation, represents a firm's ability to perceive a dynamic environment and adjust plans in alignment with its development strategy. This capability is crucial for corporate growth. Firms often face challenges such as a lack of developmental experience, inadequate financial reserves, and the dual responsibility of contributing to both national defense and economic development. These pressures result in significant social responsibilities and resource constraints. Additionally, the development of military-civilian interprises enterprises in China is characterized by complexity and high market volatility. Firms frequently struggle to promptly and accurately capture customer needs and conduct effective product development.

Further, the academic community is actively exploring the influence of entrepreneurial dynamic ability. Studies have found that environmental dynamics enhances the relationship between entrepreneurs' traits, abilities and performance [39], while environmental uncertainty positively affects the relationship between entrepreneurial orientation and performance [40]. In short, environmental dynamics amplifies the effect of resources on enterprise activities [41]. Specifically, the influence of resource bricolage behaviors, such as resource integration and utilization, on entrepreneurial performance is moderated by entrepreneurial dynamic ability [42]. When firm possess strong opportunity identification and exploitation ability, resource bricolage behaviors are more effective in enhancing corporate sustainable performance better.

## Management implications

The key development of innovative enterprises is inseparable from the participation of key innovation actors such as governments and universities. A robust social network helps enterprises acquire new technical knowledge, reduce technical costs, identify potential customers, and foster organizational management behaviors. To further enhance development, enterprises should expand their scope of collaboration, engage in cross-disciplinary learning, and establish a multi-stakeholder, cross-sector, and deeply integrated social network collaboration system [43]. Enterprises must adapt to changing times by strengthening close social ties, fostering cooperation among innovation stakeholders, and establish deep cooperation mechanisms. This approach promotes knowledge-sharing behaviors, reduce information asymmetry, and mitigates cooperation risks [44,45].

In the era of Internet and technological innovation, a firms' technological innovation ability has become increasingly important for its growth. The complex turbulent social environment places greater demands on a firm's entrepreneurial dynamic capabilities, as enterprises shoulder significant responsibilities for national development. Faced with numerous market opportunities and challenges, firms must accurately identify and evaluate the most suitable and promising entrepreneurial opportunities, helping them eliminate distractions and reduce development obstacles. Limited resources constrain the development of enterprises. Resource constraints often limit enterprise growth, and inefficiencies in resource utilization exacerbate this challenge. Resource bricolage behaviors can maximize the efficiency of existing resources by optimizing combination, enabling the development of new resources, enhancing resource interaction and circulation, and fostering innovative behaviors within the firm.

## Research limitation and future direction

This study utilized a questionnaire survey to collect data. However, the reliance on self-administered questionnaires may result in insufficient responses and compromised data authenticity. Future research could consider employing behavioral observation as an alternative to improve the reliability and validity of the data collected. Secondly, The enterprise sample for this study were sourced from various regions in China, such as eastern, western areas. Differences in economic and technological development across these regions can influence enterprise performance. To strengthen the credibility of future findings, research could control for variables such as industry type and regional economic development level. Further, this study focus on two dimensions of social networks: scale and strength. While these dimensions provide valuable insights, they are not comprehensive. Future research could investigate the management and utilization of social networks from an innovation perspective, thereby enhancing the overall value of enterprise networks. This approach would contribute to a deeper understanding of social network theory and broaden its theoretical framework.

## Author contributions

**Conceptualization:** Tianfeng Jiang, Yumeng Zhang, Haixia Feng, Hao Gao.

**Data curation:** Tianfeng Jiang, Yumeng Zhang, Haixia Feng, Hao Gao.

**Formal analysis:** Haixia Feng, Hao Gao.

**Investigation:** Tianfeng Jiang, Gang Chen, Tian Lan.

**Methodology:** Yumeng Zhang, Qian Cui.

**Resources:** Gang Chen, Qian Cui.

**Software:** Qian Cui.

**Supervision:** Haixia Feng, Yueyang Zhang.

**Validation:** Haixia Feng, Tian Lan, Yueyang Zhang.

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
