## [Decision Letter · Decision Letter 0]

PONE-D-24-11853Unveiling the Influence of Social Network Characteristics on Sustainable Performance in Small and Medium Enterprises: A Dynamic Capabilities PerspectivePLOS ONE

Dear Dr. Zhang,

Thank you for submitting your manuscript to PLOS ONE. After careful consideration, we feel that it has merit but does not fully meet PLOS ONE’s publication criteria as it currently stands. Therefore, we invite you to submit a revised version of the manuscript that addresses the points raised during the review process.

**ACADEMIC EDITOR: **

Thank you for submitting your manuscript titled "Unveiling the Influence of Social Network Characteristics on Sustainable Performance in Small and Medium Enterprises: A Dynamic Capabilities Perspective" to PLOS ONE. After careful consideration of the two reviewers' comments and a detailed review of the manuscript, I believe that the paper holds potential but requires significant revisions before it can be considered for publication.

Key Issues to Address:

Language and Readability: Both reviewers emphasized the need for thorough proofreading. The current manuscript contains numerous grammatical issues, long and convoluted sentences, and awkward phrasing. This significantly detracts from the clarity and professionalism of the paper. I recommend working with a professional language editor to revise the manuscript for readability and coherence.

Abstract and Introduction: The abstract should be revised to clearly state the study's purpose, motivation, and key findings. Similarly, the introduction lacks a clear explanation of the research motivation and the paper's contribution to the field. Please include a well-defined research question and discuss your contribution to the literature, as suggested by Reviewer 2.

Theoretical Framework: The theoretical model (Figure 1) needs further clarification. Reviewer 1 highlighted that the edges and their direction in the figure are unclear. A detailed explanation of the relationships between constructs should be added to the text. Moreover, before diving into the hypotheses, provide an overview of the theoretical analysis section for better comprehension.

Sampling and Methodology: The description of the sampling technique and the details of the questionnaire are insufficient. Reviewer 1 pointed out missing information about the sampling procedure and the questionnaire’s contents. Please provide detailed information about how the sample was selected, the representativeness of the sample, and the contents of the questionnaire. Tables detailing the sample characteristics and data collection process should be included for clarity.

Data Presentation and Analysis: Several tables, especially Table 3, are unclear and need to be restructured. Additionally, statistical results should be better explained. Reviewer 1 mentioned the need to explain model fit indices and the reliability coefficients in more detail. For instance, what does Cronbach's alpha of 0.916 imply for the study's reliability?

Literature Positioning and Contribution: As Reviewer 2 suggested, the paper's contribution to the literature needs to be articulated more clearly. This can be achieved by citing recent relevant studies (e.g., DOI: 10.1016/j.physa.2019.123174) and positioning your findings within the broader context of social network theory and resource-based theory. Clearly outline what gap in the literature this paper fills.

Once these revisions are made, I believe the paper could make a significant contribution to the understanding of how social network characteristics affect sustainable performance in SMEs. I encourage you to resubmit the manuscript with these revisions for further consideration.

We look forward to receiving your revised manuscript.

Kind regards,

Meng Xi, Ph.D.

Academic Editor

PLOS ONE

Reviewers' comments:

Reviewer's Responses to Questions

**Comments to the Author**

1. Is the manuscript technically sound, and do the data support the conclusions?

Reviewer #1: Yes

Reviewer #2: Yes

2. Has the statistical analysis been performed appropriately and rigorously? 

Reviewer #1: Yes

Reviewer #2: Yes

3. Have the authors made all data underlying the findings in their manuscript fully available?

Reviewer #1: No

Reviewer #2: Yes

4. Is the manuscript presented in an intelligible fashion and written in standard English?

Reviewer #1: No

Reviewer #2: No

5. Review Comments to the Author

Reviewer #1: Summary: In this paper, the authors propose a "feature-behavior-performance" theoretical model based on social network theory and resource-based theory. The study focuses on exploring how enterprises can obtain heterogeneous resources required for innovation through the connection with other social subjects in the network with limited resources.”

My Conclusion: I believe the manuscript is very poorly written. The grammar is not up to the mark. The sentences are unnecessarily long and hard to follow. The sampling techniques are not discussed properly, and the questionnaire details have not been revealed. So, in its current form, the paper is confusing. However, if the authors change the language and address my comments below, the paper may be considered for potential publication at Plos One.

Comments: My comments are given below.

1. The sentences are too long which makes it very hard to comprehend. For example, in Abstract: “Based on social network theory and resource-based theory, a theoretical model of social network characteristics, resource patchwork and small and medium enterprises (SMEs) sustainable performance represented by social network scale and social network strength was constructed, and 220 questionnaire data were collected.” It is advised to break down such sentences into two or even three sentences for readability.

2. In Abstract: “Using hierarchical regression, bootstrapping, structural equation model and other methods to empirically explore the impact of social network structure and relationship dimension on SMEs sustainable performance.” What does the sentence mean? Correction needed.

3. In Introduction: “Enterprise fusion eventually appeared "bubble" and "homogenization" phenomenon, interaction effect declined seriously.” What does it mean? Correction needed.

4. “In recent years, academic circles have increasingly in-depth research on enterprise development performance.” Same problem as above.

5. “. The influence of different knowledge integration, resource acquisition, risk perception and other capabilities on enterprise development.” Same problem as above.

6. The problem with the sentences, as indicated above, persists throughout the paper. The authors are strongly advised to thoroughly proofread the entire paper for such issues.

7. The abstract is a little unclear. The purpose of the study should be defined in a much clearer way.

8. The same suggestion as above for the Introduction.

9. Section: “Theoretical analysis and research hypothesis”. Before going into the subsection, an overview of the section must be provided.

10. “Based on this, this paper will explore the scale and strength of social networks from two aspects.” What are the two aspects?

11. The theoretical model presented in Fig 1 is unclear. What do the edges indicate? Is there any direction associated with the edges? Fig 1 must be explained in detail.

12. What does the following sentence mean: “In line with the key and typical characteristics of case selection.”??

13. In Sample Selection, every representative enterprise must be mentioned.

14. “First of all, understand the basic situation of the enterprise and the problems of enterprise performance.” What does it mean?

15. “During the investigation, senior leaders and staff of performance departments were asked to adjust and modify the contents of the questionnaire.” What questionnaire? What are the contents of the questionnaire? What parameters were considered when designing the questionnaire?

16. “The research team went deep into the sample enterprises to investigate.” How many sample enterprises?

17. Section “Data collection and sample situation”, paragraph 2: The statistics should be presented in the form of a table.

18. Section “Empirical Analysis”. Same suggestion as given in comment 9.

19. “Considering the situational particularity of the enterprise, the measurement of key variables is modified on the basis of the mature scale abroad, and the description and language of the scale items are modified through the pretest, so as to ensure the reliability and validity of the scale and make the research conclusions more applicable.” The sentence is way too long. Extremely hard to follow and comprehend.

20. “The scale of social network was measured with 5 item scales, and the reliability coefficient was 0.916.” What are the 5 item scales? Where are they listed? What is the definition of the reliability coefficient? What does the number 0.916 mean here? What is the scale of the reliability coefficient? What are the good, bad and neutral values of the reliability coefficient?

21. “The social network strength was measured by a 6-item scale with a reliability coefficient of 0.911.” Same questions as above.

22. The same questions for resource bricolage, entrepreneurial dynamic capability and enterprise performance.

23. Is reliability coefficient the “Cronbach's a coefficient” as indicated afterwards in Section “Reliability and validity analysis”? I think the authors refer to Cronbach's alpha coefficient. The authors are advised to give complete details of this coefficient.

24. “the mean variance variation AVE”. Do the authors mean variance extracted?

25. The authors are advised to provide full form of an acronym where it is used first (for example, KMO, SRMR, etc.).

26. The n-factor models should first be discussed in detail and then the results should be presented. In its current form, it is extremely confusing.

27. “The results showed that compared with other models, the six-factor model constructed in this study (CMIN/DF=1.791, RMSEA=0.06, SRMR = 0.054,CFI=0.925, GFI=0.817, TLI=0.917) had the best fitting indexes, and all fitting indexes met the standard.” So, what does this result entail?

28. Table 3 is a mess and cannot be understood.

29. While I understand that p is the p-value, what is r?

30. Tables need to be placed round about where they are discussed. Besides, authors are advised not to spread the table over two pages. This needs to be managed.

Reviewer #2: This article addresses the issue of social network characteristics and resource particles that affect the sustainable performance of small and medium enterprises within the framework of social network theory and resource-based theory. The study needs to be revised according to the following points:

- There are many errors in the English of the study, it should be reviewed.

- The motivation of the study should be explained in the abstract section.

- There should be a space before the references in the text. For example [11], [12], [13], [14], [16], etc.

- Some studies that have attracted attention in recent years on social networks should be cited in the introduction section. Eg, doi: 10.1016/j.physa.2019.123174, 10.1126/science.abp9364, 10.1016/j.ijinfomgt.2023.102621

- The organization of the paper should be added as a paragraph at the end of the introduction section.

- The contribution of this research to the literature should be clearly discussed in the introduction.

6. PLOS authors have the option to publish the peer review history of their article (what does this mean? ). If published, this will include your full peer review and any attached files.

**Do you want your identity to be public for this peer review?** For information about this choice, including consent withdrawal, please see our Privacy Policy .

Reviewer #1: No

Reviewer #2: No

---

## [Author Response · Author response to Decision Letter 1]

10 Dec 2024

Editor

1.Language and Readability: Both reviewers emphasized the need for thorough proofreading. The current manuscript contains numerous grammatical issues, long and convoluted sentences, and awkward phrasing. This significantly detracts from the clarity and professionalism of the paper. I recommend working with a professional language editor to revise the manuscript for readability and coherence.

We sincerely appreciate the Editor's valuable suggestions. In response to the feedback, we have engaged a professional language editor to enhance the manuscript's readability and coherence.

2.Abstract and Introduction: The abstract should be revised to clearly state the study's purpose, motivation, and key findings. Similarly, the introduction lacks a clear explanation of the research motivation and the paper's contribution to the field. Please include a well-defined research question and discuss your contribution to the literature, as suggested by Reviewer 2.

Informed by expert advice, social networks are recognized as pivotal in modern business operations. They enable enterprises to build meaningful interactions with customers, partners, and employees, gain insights into market demands and feedback, identify new collaborators, and jointly explore markets or drive innovation. Resource bricolage refers to the process by which enterprises leverage their internal strengths and resources, integrate external opportunities, and create a tailored resource system to meet their specific needs. This approach is particularly critical for resource-constrained enterprises, as it empowers them to respond flexibly to market fluctuations and environmental challenges, enhancing adaptability and innovation. Given these dynamics, exploring the interplay between social network characteristics, resource bricolage, and the sustainable performance of SMEs holds significant theoretical and practical value. This exploration forms the primary motivation for this research. The revised summary, aligned with expert recommendations, is as follows:

Gaining a deeper understanding of the mechanism through which social networks influence sustainable performance and exploring the role of resource bricolage in enhancing competitiveness can help small and medium-sized enterprises (SMEs) better adapt to market changes in the digital age and achieve sustainable development. In the context of accelerated technological change and resource scarcity, the question of how SMEs can leverage network resources to achieve sustainable growth has become a pressing issue. This study, grounded in social network theory and resource-based theory, develops a theoretical model exploring the relationships between social network characteristics, resource bricolage, and SME sustainable performance. Using survey data from 230 SMEs, the study empirically demonstrates that both social network scale and social network strength have significant positive impacts on sustainable performance. Additionally, resource bricolage mediates the effects of social network scale and strength on SME performance. Moreover, opportunity identification and opportunity exploitation—key dimensions of dynamic capabilities—moderate the relationship between resource bricolage and sustainable performance. This study highlights the mechanisms through which SMEs can leverage social networks to enhance their sustainable performance, offering theoretical guidance for effectively identifying and utilizing network resources in uncertain and dynamic environments.

Contribution

The contributions of this study are as follows: Based on resource-based theory and social network theory, this research explores the relationship between social network structure, resource bricolage, and sustainable performance. It also tests the boundary effects of dynamic capabilities on the relationship between resource bricolage and sustainable performance, thereby expanding the theoretical applicability in different contexts. This study strongly aligns with the perspective proposed by scholar Wang ( ), which emphasizes the importance of network resources in sustaining a firm's competitive advantage.。

3.Theoretical Framework: The theoretical model (Figure 1) needs further clarification. Reviewer 1 highlighted that the edges and their direction in the figure are unclear. A detailed explanation of the relationships between constructs should be added to the text. Moreover, before diving into the hypotheses, provide an overview of the theoretical analysis section for better comprehension.

We sincerely thank the experts for their valuable suggestions on improving the content of this article. In response to the recommendations provided by Reviewer 1, we have revised the theoretical framework diagram by refining the variable boundaries and using arrows to clearly indicate the directions of relationships. Additionally, we have incorporated explanations of the relationships between constructs into the main text. The detailed explanation is as follows:

On the left side of the diagram are the independent variables, representing the characteristics of social networks, which include two dimensions: social network scale and social network strength. The middle section depicts the mediating variable, representing the degree of resource bricolage. On the right side are the dependent variables, representing organizational sustainable performance. At the top of the diagram are the moderating variables, representing dynamic capabilities, which consist of two dimensions: opportunity identification capability and opportunity exploitation capability.

Finally, before examining the hypothesis, we provide an overview of the theoretical analysis section, which is as follows:

Our study, based on social network theory and resource-based theory, develops a theoretical model linking social network characteristics with sustainable performance. It also tests the mediating role of resource bricolage and the moderating effect of dynamic capabilities. The proposed theoretical hypotheses are as follows

4.Sampling and Methodology: The description of the sampling technique and the details of the questionnaire are insufficient. Reviewer 1 pointed out missing information about the sampling procedure and the questionnaire’s contents. Please provide detailed information about how the sample was selected, the representativeness of the sample, and the contents of the questionnaire. Tables detailing the sample characteristics and data collection process should be included for clarity.

We are very sorry for the insufficient description of Sampling techniques and questionnaire details in the Sampling and Methodology section. Based on this, we added the description of sampling techniques and questionnaire details, including sample selection process, sample representativeness and questionnaire content. Description is as follows :

Before the survey, the research group explained the contents and objetctives of the survey to the companies. After obtaining their consent , the research group distributed questionnaires to the senior executives and staff from the performance departments of the participating companies. The questionnaire consists of two parts. The first part focus on demographic information, including gender, age, educational background, enterprise size and years of establishment, and its influence on enterprise performance. The second part examed variables such as the characteristics of enterprise social network, resource bricolage, sustainable performance and dynamic capability (See Table 1 for details on variables). A total of 230 companies participated in the survey, Out of 300 questionnaires distributred, 220 were retrieved, and the response rate reached 73.3%.

5.Data Presentation and Analysis: Several tables, especially Table 3, are unclear and need to be restructured. Additionally, statistical results should be better explained. Reviewer 1 mentioned the need to explain model fit indices and the reliability coefficients in more detail. For instance, what does Cronbach's alpha of 0.916 imply for the study's reliability?

To ensure clarity and precision, we have revised all the charts and conducted a thorough analysis of the statistical results, including model fit indices and reliability and validity coefficients as suggested by Reviewer 1. All the charts have been presented in the final manuscript, and the statistical results are explained below.

Considering the contextual specificity of enterprise, the study modifies the measurement of key variables based on well-established scale from interntional literature. Pre-testing is conducted to refine the descriptions and language of the scale items, ensuring reliability and validity, thereby enhancing the applicability of the research findings. The questionnaire employs a seven-point Likert scale for responses, with specific details as follows.

Social Network Characteristics: Based on the views of Watson [29] and Tan [30], as well as the work of Luo and Park et al. [31], two representative dimensions, social network scale and social network strength are selected for measurement. The scale of social network consists of five items, with a Cronbach's α of 0.916. The social network strength was measured using six items, with a Cronbach's α of 0.911. Experts generally consider internal reliability insufficient if Cronbach's α is below 0.6, acceptable when it falls between 0.7 and 0.8, and strong when it ranges from 0.8 to 0.9. These resuts indicates a high reliability for themeasurement of social network characteristcs.

Resource Bricolage: by referring to the research of Davidsson et al. [32], the resource bricolage scale includs six items that assess the integration of existing resources to address with challenges and leverage opportunities to solve new problems. The scale achieves a Cronbach's α of 0.915�indicating high reliability.

Entrepreneurial Dyamic Capability: Based on the research of Ma et al. [33] and Foss et al. [34], focusing on opportunity recognition capability encompass the ability to quickly gather entrepreneurial opportunity information and identify the impact of new information,, with a scale reliability of Cronbach's α= 0.891. Opportunity exploitation ability includes six aspects, such as the ability to develop new products and services, the ability to develop new market areas, with a Cronbach's α of 0.895. Both the opportunity recognition and exploitation capability demonstrate good reliability.

Sustainable Performance: Based on the research of Zhu and Fei [35], to measure sustainable performance,,, which mainly includes enterprise growth performance and profitability performance. Items include metrics such as employees growth and new products and services growth. The Cronbach's α of this scale is 0.928, indicating that it has good reliability.

5.Literature Positioning and Contribution: As Reviewer 2 suggested, the paper's contribution to the literature needs to be articulated more clearly. This can be achieved by citing recent relevant studies (e.g., DOI: 10.1016/j.physa.2019.123174) and positioning your findings within the broader context of social network theory and resource-based theory. Clearly outline what gap in the literature this paper fills.

Thanks very much for the experts' opinions. In view of the background research of social network theory and resource-based theory, with reference to the latest relevant research, we quoted the latest relevant literature, strengthened the introduction of theoretical contributions in the introduction, and filled in some research gaps. The theoretical contributions in the introduction are as follows:

The contributions of this study are as follows: Based on resource-based theory and social network theory, this research explores the relationship between social network structure, resource bricolage, and sustainable performance. It also tests the boundary effects of dynamic capabilities on the relationship between resource bricolage and sustainable performance, thereby expanding the theoretical applicability in different contexts. This study strongly aligns with the perspective proposed by scholar Wang ( ), which emphasizes the importance of network resources in sustaining a firm's competitive advantage.

Reviewer 1:

1.The sentences are too long which makes it very hard to comprehend. For example, in Abstract: “Based on social network theory and resource-based theory, a theoretical model of social network characteristics, resource patchwork and small and medium enterprises (SMEs) sustainable performance represented by social network scale and social network strength was constructed, and 220 questionnaire data were collected.” It is advised to break down such sentences into two or even three sentences for readability.

We sincerely thank the experts for their valuable suggestions. In response, the authors have thoroughly revised the abstract and introduction sections, adjusting and optimizing sentence structures to enhance readability.The theoretical contributions in the introduction are as follows:

Abstract: Gaining a deeper understanding of the mechanism through which social networks influence sustainable performance and exploring the role of resource bricolage in enhancing competitiveness can help small and medium-sized enterprises (SMEs) better adapt to market changes in the digital age and achieve sustainable development. In the context of accelerated technological change and resource scarcity, the question of how SMEs can leverage network resources to achieve sustainable growth has become a pressing issue. This study, grounded in social network theory and resource-based theory, develops a theoretical model exploring the relationships between social network characteristics, resource bricolage, and SME sustainable performance. Using survey data from 230 SMEs, the study empirically demonstrates that both social network scale and social network strength have significant positive impacts on sustainable performance. Additionally, resource bricolage mediates the effects of social network scale and strength on SME performance. Moreover, opportunity identification and opportunity exploitation—key dimensions of dynamic capabilities—moderate the relationship between resource bricolage and sustainable performance. This study highlights the mechanisms through which SMEs can leverage social networks to enhance their sustainable performance, offering theoretical guidance for effectively identifying and utilizing network resources in uncertain and dynamic environments.

Introduction: Social networks play a crucial role in the modern business operations of enterprises. Through social networks, businesses can establish deep interactions with customers, partners, and employees, gain insights into market demands and feedback, identify new partners, and collaborate on market expansion or innovation. The diversified information exchange platforms provided by social networks facilitate the acquisition and sharing of knowledge and information, thereby driving innovation. However, social networks also present certain challenges and risks, such as the rapid spread of information, the ease with which negative information can proliferate, and the difficulty in managing online reputations.The healthy development of enterprises as an effective lever for driving the transformation and upgrading of economic structure. To achieve this goal, pursuing high-tech and high value-added industries is undoubtedly essential. However, issues such as deviations in enterprise integration concepts, insufficient innovation drive, low levels of integration among key plays, limited military-civilian techology transfer, and inadquate innovation resources significantly hider the long-term development of enterprises [1], These challenges undermine the industrial foundation necessary to support enterprise growth. Consequently, enterprises experience phenomena such as "bubblelization" and "homogenization", with a severe decline in the interactive efects among stakeholders. In order to mitigate the negative impact of resources scarcity on the enterprise growth, it is crucial for enterprises to explore rational pathway for resource acquisition.

In recent years, academic research on enterprise deve

---

## [Decision Letter · Decision Letter 1]

PONE-D-24-11853R1Unveiling the Influence of Social Network Characteristics on Sustainable Performance in Small and Medium-Sized Enterprises : A Dynamic Capabilities PerspectivePLOS ONE

Dear Dr. Zhang,

Thank you for submitting your manuscript to PLOS ONE. After careful consideration, we feel that it has merit but does not fully meet PLOS ONE’s publication criteria as it currently stands. Therefore, we invite you to submit a revised version of the manuscript that addresses the points raised during the review process.

We look forward to receiving your revised manuscript.

Kind regards,

Meng Xi, Ph.D.

Academic Editor

PLOS ONE

Journal Requirements:

Reviewers' comments:

Reviewer's Responses to Questions

**Comments to the Author**

1. If the authors have adequately addressed your comments raised in a previous round of review and you feel that this manuscript is now acceptable for publication, you may indicate that here to bypass the “Comments to the Author” section, enter your conflict of interest statement in the “Confidential to Editor” section, and submit your "Accept" recommendation.

Reviewer #1: (No Response)

Reviewer #2: All comments have been addressed

2. Is the manuscript technically sound, and do the data support the conclusions?

Reviewer #1: Yes

Reviewer #2: Yes

3. Has the statistical analysis been performed appropriately and rigorously? 

Reviewer #1: Yes

Reviewer #2: Yes

4. Have the authors made all data underlying the findings in their manuscript fully available?

Reviewer #1: Yes

Reviewer #2: Yes

5. Is the manuscript presented in an intelligible fashion and written in standard English?

Reviewer #1: Yes

Reviewer #2: Yes

6. Review Comments to the Author

Reviewer #1: Overall, the authors have largely revised the paper and according to my suggestions and comments. There still are some persisting and new issues (minor) which the authors should address. My comments to address these issues are given below:

1. In Abstract, the first sentence is again too long to comprehend. Again, it is advised to break the sentence down. The authors are also advised to do this throughout the paper wherever applicable.

2. In Introduction, last paragraph, when the authors write, “The second section…”, the authors should instead write something like this: “The rest of the paper is organized as follows. Section “Theoretical Background and Hypotheses Development” provides a theoretical analysis and proposes research hypotheses.”

3. The authors must add the contributions of the study in bullet points in Introduction. Possibly before the last paragraph in Introduction.

4. In “Theoretical Background and Hypotheses Development”, the last sentence of paragraph one, must be changed to something like, “In the subsequent sections, we discuss the proposed theoretical hypotheses.”

5. “Based on this, this study will explore the size and strength of social networks from two aspects.” Again, what are the two aspects? The authors in their response seem to suggest the two aspects are size and strength of social networks. So, the sentence must be changed accordingly. Presently, the sentence creates an unnecessary confusion.

Reviewer #2: The authors completed the revision by taking into account all suggestions. The article is acceptable as it is

7. PLOS authors have the option to publish the peer review history of their article (what does this mean? ). If published, this will include your full peer review and any attached files.

**Do you want your identity to be public for this peer review?** For information about this choice, including consent withdrawal, please see our Privacy Policy .

Reviewer #1: No

Reviewer #2: No

---

## [Author Response · Author response to Decision Letter 2]

29 Apr 2025

Reviewer 1:

Reviewer #1: Overall, the authors have largely revised the paper and according to my suggestions and comments. There still are some persisting and new issues (minor) which the authors should address. My comments to address these issues are given below:

1.In Abstract, the first sentence is again too long to comprehend. Again, it is advised to break the sentence down. The authors are also advised to do this throughout the paper wherever applicable.

According to the expert's opinion, we changed the sentence as follows:

Gaining a deeper understanding of the mechanism through which social networks influence sustainable performance and exploring the role of resource bricolage in enhancing competitiveness are valuable. These insights can help small and medium-sized enterprises (SMEs) better adapt to market changes in the digital age and ultimately, this can enable SMEs to achieve sustainable development.

Similarly, we modified other sentences with similar problems in the article:

Original text: Under the influence of technological barriers, enterprises face obstacles in the process of technological development due to limited innovation resources. Innovation resources in enterprises ten to evolve rapidly, with some resources depreciating quickly and being non-renewable, lending to inefficiency and resource waste [11].

Revised sentence: Technological barriers often hinder enterprises' technological development due to limited innovation resources. These resources evolve rapidly, with some depreciating quickly or becoming non-renewable. As a result, enterprises face inefficiencies and resource waste, which further constrain their innovation capabilities.

Original text: In addition, entrepreneurial dynamic capabilities enhance and organization’s sensitivity to fluctuations in the external environment, enabling the integration of strategic planning capabilities to address environmental changes and promptly adjust production and operational strategies [30].

Revised sentence: Furthermore, entrepreneurial dynamic capabilities strengthen an organization’s ability to detect and respond to external environmental fluctuations. By integrating strategic planning capabilities, organizations can effectively adapt to environmental changes and quickly adjust their production and operational strategies.

Original text: Taking social network scale and strength as representative characteristics of social network structure, the empirical research finds that an expaned social network scale - characterized by broader business scope and diverse service targets - stimulate the demand for resource richness and promotes resource bricolage behaviors.

Revised sentence:The empirical research uses social network scale and strength as key indicators of social network structure. It finds that an expanded social network scale—reflected by a broader business scope and diverse service targets—increases the demand for resource richness. This, in turn, encourages resource bricolage behaviors.

2.In Introduction, last paragraph, when the authors write, “The second section…”, the authors should instead write something like this: “The rest of the paper is organized as follows. Section “Theoretical Background and Hypotheses Development” provides a theoretical analysis and proposes research hypotheses.”

Thanks to the valuable advice of the experts, we have rearranged the last paragraph of the introduction, which is as follows:

The rest of the paper is organized as follows. Section “Theoretical Background and Hypotheses Development” provides a theoretical analysis and proposes research hypotheses. Section "Methodology" introduces the source of sample data, the measurement process of variables, etc., and analyzes the data results in detail. Section “Discussion and Conclusion” discusses the research findings and offers managerial implications.

3. The authors must add the contributions of the study in bullet points in Introduction. Possibly before the last paragraph in Introduction.

Thanks very much for the experts' opinions. In view of the background research of social network theory and resource-based theory, with reference to the latest relevant research, we quoted the latest relevant literature, strengthened the introduction of theoretical contributions in the introduction, and filled in some research gaps. The theoretical contributions in the introduction are as follows:

From the perspective of current research, the research on social network and enterprise performance is relatively abundant [19]. However, due to the plight of small and medium-sized enterprises (SMEs) facing threats to their survival due to insufficient resources [20], there is still a gap in the research on social network and sustainable performance of SMEs with SMEs as the main body of research. In addition, from the perspective of resource patchwork, the existing literature has not been answered to explore the mechanism of social network and sustainable performance.The contributions of this study are as follows: Based on resource-based theory and social network theory, this research explores the relationship between social network structure, resource bricolage, and sustainable performance. It also tests the boundary effects of dynamic capabilities on the relationship between resource bricolage and sustainable performance, thereby expanding the theoretical applicability in different contexts. This study strongly aligns with the perspective proposed by scholar Wang [21], which emphasizes the importance of network resources in sustaining a firm's competitive advantage.

4.In “Theoretical Background and Hypotheses Development”, the last sentence of paragraph one, must be changed to something like, “In the subsequent sections, we discuss the proposed theoretical hypotheses.”

According to the expert's opinion, we changed the sentence as follows:“In the subsequent sections, we discuss the proposed theoretical hypotheses”.

5.“Based on this, this study will explore the size and strength of social networks from two aspects.” Again, what are the two aspects? The authors in their response seem to suggest the two aspects are size and strength of social networks. So, the sentence must be changed accordingly. Presently, the sentence creates an unnecessary confusion.

Thank you very much for the expert opinion, the two aspects are size and strength of social networks. According to the expert's opinion, we changed the sentence as follows:

“Based on this, this study will explore the size and strength of social networks”.

---

## [Editor Report · Decision Letter 2]

Unveiling the Influence of Social Network Characteristics on Sustainable Performance in Small and Medium-Sized Enterprises : A Dynamic Capabilities Perspective

PONE-D-24-11853R2

Dear Dr. Feng,

We’re pleased to inform you that your manuscript has been judged scientifically suitable for publication and will be formally accepted for publication once it meets all outstanding technical requirements.

Kind regards,

Meng Xi, Ph.D.

Academic Editor

PLOS ONE
---

## [Editor Report · Acceptance letter]

PONE-D-24-11853R2

PLOS ONE

Dear Dr. Feng,

I'm pleased to inform you that your manuscript has been deemed suitable for publication in PLOS ONE. Congratulations! Your manuscript is now being handed over to our production team.

Kind regards,

on behalf of

Professor Meng Xi

Academic Editor

PLOS ONE